# Daily Practice Assessment of *KRAS* Status in NSCLC Patients: A New Challenge for the Thoracic Pathologist Is Right around the Corner

**DOI:** 10.3390/cancers14071628

**Published:** 2022-03-23

**Authors:** Christophe Bontoux, Véronique Hofman, Patrick Brest, Marius Ilié, Baharia Mograbi, Paul Hofman

**Affiliations:** 1Laboratory of Clinical and Experimental Pathology, Pasteur Hospital, Centre Hospitalier Universitaire de Nice, Université Côte d’Azur, BB-Biobank 0033-0025, 06002 Nice, France; bontoux.c@chu-nice.fr (C.B.); hofman.v@chu-nice.fr (V.H.); ilie.m@chu-nice.fr (M.I.); 2FHU OncoAge, Biobank BB-0033-00025, Université Côte d’Azur, CHU Nice, 06001 Nice, France; patrick.brest@unice.fr (P.B.); baharia.mograbi@unice.fr (B.M.); 3IRCAN Team 4, Inserm U1082/CNRS 7284, Centre de Lutte Contre le Cancer Antoine Lacassagne, 06000 Nice, France

**Keywords:** lung cancer, *KRAS*, molecular biology, algorithms, samples

## Abstract

**Simple Summary:**

*RAS* mutation is the most frequent oncogenic alteration in human cancers and *KRAS* is the most frequently mutated, notably in non-small cell lung carcinomas (NSCLC). Various attempts to inhibit *KRAS* in the past were unsuccessful in these latter tumors. However, recently, several small molecules (AMG510, MRTX849, JNJ-74699157, and LY3499446) have been developed to specifically target *KRAS* G12C-mutated tumors, which seems promising for patient treatment and should soon be administered in daily practice for non-squamous (NS)-NSCLC. In this context, it will be mandatory to systematically assess the *KRAS* status in routine clinical practice, at least in advanced NS-NSCLC, leading to new challenges for thoracic oncologists.

**Abstract:**

*KRAS* mutations are among the most frequent genomic alterations identified in non-squamous non-small cell lung carcinomas (NS-NSCLC), notably in lung adenocarcinomas. In most cases, these mutations are mutually exclusive, with different genomic alterations currently known to be sensitive to therapies targeting *EGFR*, *ALK*, *BRAF*, *ROS1*, and *NTRK*. Recently, several promising clinical trials targeting *KRAS* mutations, particularly for *KRAS G12C*-mutated NSCLC, have established new hope for better treatment of patients. In parallel, other studies have shown that NSCLC harboring co-mutations in *KRAS* and *STK11* or *KEAP1* have demonstrated primary resistance to immune checkpoint inhibitors. Thus, the assessment of the *KRAS* status in advanced-stage NS-NSCLC has become essential to setting up an optimal therapeutic strategy in these patients. This stimulated the development of new algorithms for the management of NSCLC samples in pathology laboratories and conditioned reorganization of optimal health care of lung cancer patients by the thoracic pathologists. This review addresses the recent data concerning the detection of *KRAS* mutations in NSCLC and focuses on the new challenges facing pathologists in daily practice for *KRAS* status assessment.

## 1. Introduction

The most recent global report on the epidemiology of cancer diseases states that lung cancer has the highest mortality among 36 cancer types considered, and it is the second most frequently diagnosed cancer type around the world [1]. Therefore, in 2020, based on data collected from 185 countries, the number of diagnosed cases was estimated at more than 2.2 million cases (11.4% of all cancers), while mortality was around 1.8 million cases (18.0%) [1,2]. It is well known that the mortality of NSCLC is strongly associated with late diagnosis [1,3,4]. Among the lung cancer histological subtypes, non-small cell lung carcinoma (NSCLC) is the most frequent histological subtype, representing at least 85% of all cases [5]. The therapeutic strategy of advanced-stage or metastatic NSCLC, notably of non-squamous (NS)-NSCLC, has evolved dramatically in the last few years, thanks to the development of new targeted therapies and immune checkpoint inhibitors (ICIs) administered alone or in association with chemotherapy and targeted therapies [6,7,8]. These different therapies have significantly increased the overall survival of NS-NSCLC patients [9,10]. Therefore, several therapeutic molecules for routine clinical practice were rapidly granted marketing authorization by the Food and Drug Administration (FDA) in the US and by the European Medicines Administration (EMA) in Europe [11,12,13]. Different drugs target genomic alterations in *EGFR*, *ALK*, *ROS1*, *BRAF*, and *NTRK*, or more recently in *RET* and *MET* [11,12,13]. In the case of absence of oncogenetic addiction, ICIs are now widely used as significant alternative therapeutic options [7,8]. Moreover, besides the different promising ICIs, new molecules are currently being evaluated in many clinical trials and recent results suggest the coming soon administration of additional targeted therapies for NS-NSCLC in daily practice [10,12,14,15,16,17]. Among these new drugs, some target *KRAS* mutations, particularly the *KRAS* G12C mutation [18,19,20,21]. The development of these therapies raises a lot of hope and expectations in thoracic oncology since the *KRAS* mutations, which have been considered for a long time as non-targetable, are identified in more than 30% of NSCLC patients, at least in Europe and in the US, depending on the different populations [15,22,23,24,25,26,27]. However, among the *KRAS*-mutated NSCLC, a *KRAS* G12C mutation has been identified in 30% to 50% of cases and currently represents the only targetable mutation associated with different clinical trials or allowing the treatment of patients with one of the KRAS G12C inhibitors (sotorasib), which has been recently approved by the FDA/EMA for second line treatment based on phase II results (CodeBreak 100 clinical trial) [23,25,26,28,29,30]. It is noteworthy that other different solid tumors can have a *KRAS* mutation, notably a *KRAS* G12C mutation, such as some colon and pancreatic carcinomas [31,32]. In this regard, a recent clinical trial [CodeBreaK100 (NCT03600883)] demonstrated the efficacy of the sotorasib in pancreatic carcinoma mutated for *KRAS* G12C [33].

*KRAS* has been added to the list of genes that certainly must be characterized for advanced-stage NS-NSCLC at diagnosis, progression, and recurrence [8]. One of the consequences is a modification of the diagnostic algorithms used by most of the clinical and molecular pathology laboratories. So, the challenge will now be to set up and use specific and sensitive molecular testing for *KRAS* status assessment, considering notably both the quality and quantity of the DNA that is extracted from tissue, cytological, and blood samples, but also the turnaround time (TAT) to send the report to the physician.

After a brief reminder of the recent data concerning the biology and epidemiology of *KRAS* mutations in NSCLC, notably of the *KRAS* G12C mutation, as well as their impact on patient prognosis, this review reports the strategic issues and new challenges faced in the assessment of the *KRAS* status in daily practice by the thoracic pathologists.

## 2. KRAS Biology and the Signaling Pathways Induced by *KRAS* Mutations

A substantial number of studies related to the biology of KRAS have been performed and are being published almost every day, notably into lung cancer, highlighting the fact that this topic represents a huge stake in thoracic oncology. We report here some of the major and specific knowledge in the field, mainly considering the signaling pathways induced in lung cancer cells by the *KRAS* mutation. *KRAS* mutations affect cell biology since, in a non-mutated *KRAS* cell, the KRAS protein, which is a small guanosine triphosphate (GTPase), connects cell membrane growth factor receptors to intracellular signaling pathways and many transcription factors, thus playing a pivotal role in a variety of cellular processes [34,35,36]. The *KRAS* gene produces two isoforms (KRAS-4A and KRAS-4B), each of which has a distinct C-terminus and hence might cause various membrane interactions within different lipid raft interactions. The two isoforms coexist in cells, and although KRAS-4B is the most common, both can cause lung cancer in mice. However, even in the absence of the KRAS-4B isoform, KRAS-4A has been shown to cause metastatic lung adenocarcinomas in vivo [37]. The binding of ligands to their transmembrane receptors, usually receptor tyrosine kinases, activates the upstream signaling pathways of KRAS, as does recruitment of docking proteins, such as GRB2, in association with RAS-specific GEFs, which enhance KRAS activation. Of note, only certain *KRAS* mutant lung cancers (those expressing both galectin-3 and integrin αvβ3) rely on a process of anchorage-independent growth, showing the close relationship between membrane signaling and KRAS dependence in lung cancer [38,39]. Just upstream of KRAS, SOS1 is a guanine exchange factor (GEF) that binds and activates the GDP-bound RAS family of proteins at its catalytic binding site, promoting GDP to GTP exchange. In addition to its catalytic site, SOS1 can engage GTP-bound KRAS at an allosteric site that enhances its GEF function. SOS1 depletion or specific genetic inactivation of its GEF function has been shown to reduce the survival of *KRAS*-mutated tumor cells [40,41].

KRAS downstream signaling has been extensively investigated over recent decades and depends on the tissue of origin [42,43]. Four major routes downstream of KRAS have been characterized: the RAF/MEK/ERK MAPK, PI3K/AKT, RALGEF/TBK1 and RAF1-MAPK independent pathways. The RAF/MEK MAPK pathway activates transcription factors via ERK nuclear translocation to stimulate the expression of diverse genes involved in cell proliferation, survival, differentiation, and cell-cycle regulation. While this pathway is normally activated in response to growth factors, mutations in *EGFR, KRAS*, and *BRAF* maintain a residual activity and drive oncogenic addiction. The PI3K/AKT pathway also plays a key role in RAS-mediated tumorigenesis, in which AKT phosphorylation leads to the activation of major downstream targets such as the mammalian target of rapamycin (mTOR), forkhead box O (FOXO), or nuclear factor (NF)-κB, which stimulate survival, cell-cycle progression, metabolism, and invasion. While studied mainly in other KRAS dependent cancers, the importance of the RAL/TBK1 pathway in RAS-induced lung cancer was confirmed in an RNAi screen to identify lethal partners of oncogenic *KRAS* [44]. Given the importance of tumor immunogenicity in the response to ICIs, TBK1 seems to play a central role since it contributes to innate immunity by activating interferon regulatory factor 3/7 (IRF3/7), thereby inducing type 1 interferon gene expression and supporting cell growth and self-renewal [45,46,47]. The unexpected requirement for RAF1 in KRAS-driven NSCLC has been identified in mouse models as a new and important independent pathway. RAF1 has been shown to be important for the initiation and progression of NSCLC [48]. Interestingly, elimination of RAF1 expression led to a drastic reduction in lung tumors without affecting the activity of the MAPK pathways, illustrating a new pivotal downstream route of regulation. c-RAF ablation induced regression of advanced Kras/Trp53 mutant lung adenocarcinoma by a mechanism independent of MAPK signaling [48]. Of note, the four non-exhaustive pathways cited above can be activated differently depending on the mutation, either heterozygous or homozygous, or on the expressed KRAS isoform, in synergy with the *KRAS* wild-type allele and other RAS family members, i.e., NRAS and HRAS [49,50,51]. Therefore, even if a lot of knowledge has been obtained over the last few years, further research is still needed to define a comprehensive scenario of the context of activation of downstream pathways of KRAS in lung tumors.

The depletion or accumulation of different metabolites, which results from aberrant cancer metabolism, has different ramifications for the function of surrounding immune cells, and thus has an impact on the tissue microenvironment. To manage the metabolic challenges imposed by this microenvironment, cancer cells can participate in many cooperative metabolic interactions that support tumor growth and invasion, as well as competitive metabolic interactions that induce tumor survival by limiting nutrient availability to the antitumor immune system [34,52]. Taken together, increasing our knowledge of the different effector pathways downstream of KRAS, such as dysregulated metabolism, might lead in the future to the enhancement of many therapeutic interventions both alone and in combination with direct inhibition of KRAS. Additionally, a recent study from Kobayashi and colleagues nicely provides new insights into the biological role of silent mutations in *KRAS* and their potential to be translated into novel therapies [53]. Briefly, synonymous single-nucleotide polymorphisms (sSNPs) that alter programmed translational speed affect expression and function of the encoded protein [54,55]. The sSNPs do not result in the change of the amino acid sequence because these nucleotide changes usually occur in the third base of a codon. This often leads to the conclusion that a lack of protein sequence alteration will not have any functional consequences [54,56,57]. However, sSNPs can affect the expression of neighboring genes, but also the mRNA splicing, stability, andstructure, as well as protein function and folding. Synergistic advances in next-generation sequencing have led to the identification of sSNPs associated with disease penetrance [55]. Therefore, over 50 human non-tumoral and tumoral diseases have been associated with these synonymous mutations [54,56,57]. So, it is noteworthy that these types of mutations in *KRAS* were proven to be crucial for protein and mRNA expression and thus can lead to amplification and overexpression of this gene. As a result of this sSNP, it was demonstrated that the cell proliferation and metastasis can be enhanced, and the resistance to targeted therapeutics can be increased [53]. This discovery presents an opportunity for new treatments targeting *KRAS* in NSCLC.

## 3. Epidemiology of *KRAS* Mutations in Non-Small Cell Lung Carcinoma

Depending on the population and the series, the frequency of *KRAS* mutations in lung cancer patients is variable, identified in around 20% to 40% of lung adenocarcinomas and in around 5% to 7% of lung squamous cell lung carcinomas [58,59,60,61,62,63,64,65]. These mutations are much more frequent in Caucasian and Hispanic populations (from 25% to 35%) than in Asian populations (including Chinese and Indian patients) and in Iranians (globally from 9% to 13%) [59,61,62,63,65,66,67,68]. Of interest, a recent study identified a *KRAS* mutation in 24% of NSCLC Chinese patients [69]. *KRAS* mutations are more frequent in smokers (30%) than in non-smoker (10%) populations [61,62,63,65,66,67,68]. Other epidemiological factors have been associated with these mutations (gender, age, histological subtypes of lung adenocarcinoma, histological grading, tumor stage), leading to some discrepant results [60,62,63,70,71,72,73].

Different *KRAS* mutation subtypes have been defined in NSCLC [60]. *KRAS* mutations occur in the GTP-binding domain of the protein, mainly affecting codons 12 and 13 in exon 2 and codon 61 in exon 3 [64]. However, the majority of these mutations are substitutions of glycine in codon 12: G12C (40−50% of all *KRAS* mutations), G12V (11−26%), G12D (6–17%), G12A (6–12%), and G12S (2–5%) [60,62,74,75,76]. Other *KRAS* mutations include different rare mutations in codon 12 [G12R (1.7%)], codon 13 [G13C (6–12%); G13D (3–9%)], and codon 61 [Q61L (1–2%), Q61H (4%)]. Exceptional mutations in exon 4 affecting codon 146 have also been reported in less than 1% of cases, notably in metastatic NS-NSCLC. *KRAS* G12C and *KRAS* G12V are more frequent in smokers while *KRAS* G12D and *KRAS* G12S are more frequent in non-smoker patients [77]. The proportion of *KRAS* G12C-mutated subtypes is slightly lower across a series of Asian patients in comparison with Caucasian patients [61].

*KRAS* mutations can occur as a result of NSCLC progression treated with different tyrosine kinase inhibitors (TKIs) [78,79]. Therefore, a *KRAS* G12C mutation is noted in around 1% of patients showing tumor progression after first-line treatment with TKIs [79].

## 4. Impact of the *KRAS* Status on the Prognosis of Non-Small Cell Lung Carcinoma Patients

It is noteworthy that many contradictory studies into the prognostic value of *KRAS* mutations in NSCLC patients have been published [80,81,82]. Therefore, when examining these studies and drawing conclusions, the prognostic role of a *KRAS* mutation in lung cancer is still debatable. This is probably firstly due to the different methodologies used for molecular testing (notably the sensitivity and specificity of the different approaches), but also to variable patient inclusion criteria (including age, gender, and ethnicity) to assess the prognostic impact of these *KRAS* mutations. Moreover, the prognosis of *KRAS*-mutated NS-NSCLC can vary depending on the disease stage [83,84,85]. Recent studies have shown a higher risk of tumor recurrence after surgical resection in early-stage cancers with *KRAS* G12C mutations in comparison to other *KRAS*-mutated tumors and non-*KRAS*-mutated tumors [83,84,85]. Thus, it is certainly of strong interest to look systematically for the *KRAS* mutational status in resected lung specimens. 

Some studies integrated the relationship between the *KRAS* mutation and other oncogenic mutations, the PD-L1 expression, and/or the tumor mutational burden (TMB) [74,86,87]. These different biological criteria can certainly have a strong impact on the behavior of *KRAS*-mutated tumors. Thus, according to their evaluation, this may explain some discrepancies among studies related to the impact of *KRAS* mutations on disease prognosis. Therefore, some initial series only evaluated the *KRAS* status without looking for different associated genomic alterations, notably in *STK11*, *KEAP1*, and/or *TP53* [88]. However, more than 50% of the *KRAS* mutated-NSCLC show a concurrent genomic alteration on one or more other genes (mainly, *TP53, STK11, KEAP1, CDKN2A, AKT1, PI3KA, BRAF*) [89,90,91]. The most common co-mutations are observed in *TP53* (35–40%) and *STK11* (12–20%) [89,90,91,92]. The prognosis of NS-NSCLC patients may also vary according to the subtype of the *KRAS* mutation. Therefore, the prognostic impact of *KRAS* G12C is debatable in certain series [83,84,85,93]. A recent study showed that *KRAS* G12C is mutually exclusive compared to known actionable driver mutations, but non-driver co-mutations are common (*STK11*, 21.5%; *KEAP1*, 7.0%; *TP53*, 48.0%) [94]. *KRAS* G12C mutations can also be associated with *ERBB2* amplifications and *ERBB4* mutations [73]. *KRAS* G12V mutations are frequently associated with *PTEN* mutations, while *KRAS* G12D mutations are associated with *PDGFRA* mutations [90]. *TP53* mutations on certain exons, such as exon 8, have been associated with worse prognosis in certain series of NSCLC patients [95]. Moreover, prognosis may even be worse in the case of *TP53, STK11*, and *KEAP1* co-mutations [59,63,76,82,88,96,97,98,99,100,101,102,103,104,105]. An experimental study demonstrated that NRF2 activation, in association with a loss in *STK11* and *KRAS* activation, can be associated with a negative prognosis and tumor progression, and can lead to resistance to immunotherapy [106]. Globally, the presence of several co-mutations in association with the *KRAS* mutation leads to more aggressive tumors and chemo-resistance. Moreover, the association between *KRAS* mutations and a high level of gene copy number variation of *KRAS* can be an indicator of worse prognosis [107].

The prognosis of patients with *KRAS*-mutated tumors is associated with the level of PD-L1 expression in tumor cells [63,108,109,110,111,112]. Thus, *KRAS*-mutated tumors with a high level of PD-L1 expression demonstrated worse prognosis [63,108,109,110,111,112]. Moreover, some studies showed that PD-L1 expression is also linked to *TP53* mutations and tobacco exposure [113,114]. PD-L1 expression seems to be variable according to the subtype of the *KRAS* mutation [108,115]. Independently of the tobacco status, the TMB is higher in *KRAS* mutations than in wild-type *KRAS* NS-NSCLC [116]. However, the TMB level is lower in *KRAS* G12D mutated tumors than in other *KRAS*-mutated tumors [86]. This could be explained by the fact that *KRAS* G12D is most often identified in non-smoker patients [86].

## 5. The *KRAS* G12C Mutation Is a New Target for Promising Therapy in Advanced-Stage or Metastatic Non-Small Cell Lung Carcinoma

It was demonstrated several years ago that *KRAS*-mutated NSCLCs are more chemo-resistant than *KRAS* wild-type NSCLCs [117]. Moreover, *KRAS*-mutated NSCLC showed no response to TKIs targeting *EGFR* mutations [118]. So, *KRAS* mutations have been considered until recently to not be “druggable” with some targeted therapies [26,119]. Initially, different strategies focused on direct targeting of KRAS proteins and downstream inhibition of KRAS effector pathways [120,121]. However, most of these strategies were associated with a lack of tumor responsiveness [120,121]. Therefore, the identification of a hidden pocket adjacent to switch II in GDP-bound KRAS suddenly renewed great interest in the direct targeting of the mutant KRAS proteins [122]. So, different molecules were rapidly developed, in particular those targeting KRAS G12C mutated NSCLC (Figure 1) [105,106]. However, the initial KRAS G12C inhibitors were not sufficiently potent for further development for humans [105]. More recently, novel covalent small-molecule inhibitors of *KRAS* G12C mutant proteins, notably AMG 510 (sotorasib) and MRTX849 (adagrasib), provided initially promising results [123,124,125,126,127,128]. Therefore, after the positive results obtained in vitro and in vivo on inhibition of growth of different *KRAS* G12C mutated cells, new drugs are currently under evaluation on patients in different clinical trials and should soon be administered in daily practice to advanced or metastatic NS-NSCLC patients showing a *KRAS* G12C mutation [30,129,130]. Thus, promising results have been obtained in phase I and phase II clinical trials with sotorasib and adagrasib targeting *KRAS* G12C mutated NSCLC showing 37% and 45% overall response rate (ORR), respectively. In these studies, *KRAS* and *STK11* co-mutations confer a better response to targeted therapies [23,25,30,123,131,132,133,134,135]. As mentioned previously, only one drug (sotorasib) has been FDA and EMA approved as second line therapy in *KRAS* G12C mutated tumors [23,28,29]. Caution in interpretation is necessary until the advantages of overall survival of some other direct inhibitors of mutant proteins are confirmed in large-scale phase III randomized trials. In addition, many clinical trials are currently being developed to evaluate additional KRAS G12C inhibitors, such as LY3499446 (clinicaltrials.gov identifier: NTC04165031) and JNJ-746999157 (clinicaltrials.gov identifier: NTC0400630) and other *KRAS* mutation-specific inhibitors in NSCLC patients (Table 1) [136,137,138]. In fact, the different KRAS inhibitors can be classified in different therapeutic families, including *KRAS* mutation-specific inhibitors, pan-KRAS inhibitors, downstream KRAS inhibitors, upstream KRAS inhibitors, and finally, some inhibitors targeting cellular metabolism and autophagy [139].

## 6. Resistance Mechanisms Induced in Non-Small Cell Lung Carcinomas by Specific Inhibitors Targeting the *KRAS* G12C Mutation

More than 50% of the patients treated with sotorasib and adagrasib do not show significant tumor reduction when treated with these specific KRAS G12C inhibitors [140,141,142,143]. Moreover, resistance to these molecules occurred relatively early, most of the time a few months following the treatment initiation [139]. Therefore, recent clinical data showed that the duration of response for the great majority after sotorasib treatment is quite short, since the median progression-free survival is around 6 months [23]. These different results stress the urgent need to better characterize the resistance pathways that can be induced by the KRAS G12C inhibitors [139,142]. In fact, different mechanisms of primary and secondary resistances occurring in *KRAS*-mutated NSCLC patients treated with these inhibitors can explain these clinical results (Figure 2) [141,143,144,145,146,147,148,149,150,151,152,153,154,155]. Most of these resistance mechanisms are driven by genomic alterations (*KRAS* mutation or amplification or mutations on other genes) or by different bypass mechanisms [143,145,147]. So, many mechanisms of resistances are due to genetic alterations in the nucleotide exchange function or adaptive mechanisms in downstream pathways or in expressed *KRAS* G12C [143,148,156,157]. Thus, *KRAS* G12C can stay in an active state as a result of activation of the *EGFR* or *AURKA* signaling pathways [157]. This can be mediated by PTPN11/SHP2 recruiting SOS1 to promote conversion to an active state [158]. 

Primary or intrinsic resistance mechanisms are often associated with the fact that not all *KRAS* mutant cells depend on KRAS activation to maintain their viability, and this can be maintained despite ablation of the KRAS mutant protein, highlighting the heterogeneity of the different *KRAS*-mutated tumor cell subclones [159]. In this regard, it was shown that activation of the downstream effectors (i.e., ERK and AKT) was not suppressed after KRAS knockdown. The RAS–RAF–ERK pathway has multiple independent mechanisms that maintain signaling in its active state while being selectively targeted. Therefore, intrinsic resistance relies on this as a fundamental pathway for cell survival [143]. 

The predominant secondary resistance mechanisms are linked to the onset of new *KRAS* mutations [145,147]. One major new mutation, the Y96D mutation, occurs at a relevant position for sotorasib and adagrasib binding, leading to resistance to both inhibitors [141,149]. Other *KRAS* mutations, such as R86S and H95D, have been identified in association with secondary resistance to KRAS G12C inhibitors [141]. In addition, acquired mutations can be identified alone or in association with acquired mutations in the switch II pocket [141]. So, G12D, G12V, and G12W in trans can also be detected at tumor progression [141,143]. Consequently, different clones can present resistance by impeding the binding of drugs to *KRAS* G12C mutated cells and other tumor cells acquire resistance through mutations that are not targetable by KRAS G12C inhibitors. As already described for tumors rearranged for *ALK* and treated with crizotinib, amplification in the *KRAS* G12C allele can also be an independently acquired resistance mechanism to KRAS G12C inhibitors [143,145,147]. Off-target mechanisms of secondary resistance, occurring on different genes, are also possible [126,128,130]. So, in a similar manner to that described for many TKIs (notably those targeting *EGFR* mutations or *ALK* rearrangements), secondary resistance associated to KRAS G12C inhibitors can be linked to *MET* amplification [143,147,152,160,161]. A *RET* fusion can be a mechanism of secondary resistance since RET kinase domain activation leads to oncogenic signaling of the PI3K-AKT, JNKs, and BRAF–MEK–ERK pathways, thus promoting cell survival and tumor promotion [162]. Other fusions have been reported to be associated with resistance to adagrasib such as *ALK* and *FGFR3* fusions, or more rarely, *BRAF* and *NTRK1* fusions [140,143,147]. Polyclonal activation of different RAS isoforms, associated with different *NRAS* mutations (such as Q61L, Q61R, and Q61K) can play an important role in secondary mechanisms of resistance to KRAS G12C inhibitors [143,145]. Other mutations of *BRAF*, *PI3KCA*, and *PTEN* can be associated with this resistance [145,147]. In addition, an important off-target mechanism of resistance to adagrasib is histological transformation, such as squamous cell carcinoma transformation [141]. From a practical point of view, early identification of some of these different mechanisms of resistance can allow alternative treatments to NS-NSCLC harboring a *KRAS* G12C mutation to be proposed. Finally, among one of the recently identified mechanisms, BCL6 expression seems to be an important cause of resistance [163]. As consequences of these different mechanisms of resistance, many clinical trials are ongoing or are going to soon start combining a KRAS G12C inhibitor and different other therapeutic agents, some of which will later be detailed in the perspectives section [164]. Briefly, as an example, the combination therapies currently programed in the CodeBreak 101 clinical trials include the association of soratosib with different molecules such as AMG 404, trametinib, RMC-4630, afatinib, pembrolizumab, panitumumab, carboplatin, pemetrexed, and docetaxel, atezolizumab, everolimus, palbociclib, MVASI^®^ (bevacizumab-awwb), TNO155, FOLFIRI, and loperamide [164].

Abbreviations: RTK: receptor tyrosine-kinase, *KRAS:* Kirsten rat sarcoma, GAP: GTPase activating proteins, GEF: guanine nucleotide exchange factors, SOS: son of sevenless, AURKA: Aurora kinase B, SOS: son of sevenless, SHP2: Src homology region 2 domain-containing phosphatase-2, Ral: Ras-like, NF-kB: nuclear factor-kB, RAF: RAF proto-oncogene serine/threonine-protein kinase, MEK: Mitogen-activated protein kinase kinase, ERK: extracellular signal-regulated kinase.

## 7. Impact of *KRAS* Mutations on the Efficiency of Immunotherapy

Previous studies demonstrated that the outcome of advanced or metastatic NS-NSCLC patients treated with ICIs was better for tumors with *KRAS* mutations compared to the outcome with wild-type *KRAS* tumors [114]. This can probably be explained by a higher overall level of PD-L1 expression in *KRAS*-mutated tumors [87]. Therefore, PD-L1 expression is relevant for prediction of the efficacy of ICIs for patients with *KRAS*-mutated tumors more than in patients with other types of NSCLC [87,115,165,166,167]. In the study by Jeanson et al., the sensitivity of ICIs appears to be associated more with the expression of PD-L1 by the tumor cells than with the different types of *KRAS* mutations [115]. In the study by Lauko et al., the response to ICIs of NSCLC patients with a brain metastasis was higher in *KRAS*-mutated tumors [168]. In a recent study that enrolled patients with PD-L1 positive tumors, pembrolizumab monotherapy did not significantly improve the outcome of *KRAS*-mutated tumors compared with chemotherapy for patients with wild-type disease [169]. So, in addition to the *KRAS* mutational status, some authors propose to take into consideration not only the expression of PD-L1 but also other biological parameters such as the density of CD8 positive lymphocytes infiltrating the tumor cells when assessing the response to ICIs [165]. Moreover, the ICI response with or without platinum-based chemotherapy or with or without anti-CTL4 treatment is also dependent on the presence of *TP53*, *STK11*, and/or *KEAP1* co-mutations [170,171,172,173]. Thus, the association of these different genomic alterations leads to variable immune signatures [170,171,172,173,174]. Most of the studies showed that the association of *STK11* and *KEAP1* mutations with a *KRAS* mutation leads to ICIs resistance [104,174]. *KRAS* and *TP53* mutations are frequently associated with NSCLC [62,88,172]. These mutated NSCLC seem to present a long-term response to ICIs [175]. Finally, different subgroups of tumors can be distinguished, certain tumors showing a *KEAP1* mutation in the absence of a TP53 mutation lead to an immune desert, while both *KEAP1* and *TP53* associated mutations show a rich immune infiltrate [172,176]. Other genes of interest such as *SMARCA4* should also be studied, since *SMARCA4* and *KRAS* co-mutations lead to a weak response to ICIs [177].

Currently, the first-line treatment in daily practice of patients with *KRAS*-mutated tumors associates ICIs (pembrolizumab) and chemotherapy or immunotherapy (pembrolizumab) alone for tumors expressing PD-L1 in more than 50% of tumor cells [7,178]. A therapeutic strategy using specific KRAS inhibitors alone might be administered in the near future as first-line treatment for tumors expressing PD-L1 in less than 50% of tumor cells [23,30,129,130]. In this regard, a clinical trial (CodeBreak 201) has recently started in *KRAS* G12C mutated aNS-NSCLC for sotorasib administration in tumors with less than 1% of PD-L1 expression and/or STK11 mutation [179]. Association of this targeted therapy with immunotherapy could also be proposed [22]. For this treatment it is mandatory that no genomic alteration in *EGFR*, *ALK, ROS1, BRAF,* and *NTRK* is detected. 

Some studies showed that *STK11* mutations are less frequent in all *KRAS*-mutated tumors than in non *KRAS*-mutated tumors [180]. However, the *KRAS* G12C mutation can be associated with *STK11* and/or *KEAP1* mutations and show a low response to immunotherapy [89,91,92,104,181,182]. Therefore, in the case of these co-mutations, specific KRAS G12C inhibitors may be a favorable approach as first-line therapy. Another approach focused on combined treatment that can associate KRAS G12C inhibitor molecules and immunotherapy [22]. This combined therapy may shift the balance away from an immuno-suppressive tumor microenvironment to allow effective anti-tumor immunity [22,183]. In this regard, it seems highly appropriate to rapidly set up for late-stage NS-NSCLC patients reflex testing looking for the status of *KRAS, STK11*, and *KEAP1* before administrating any first-line therapy. However, we have to keep in mind that some NS-NSCLC showing *KRAS* and *STK11* mutations also respond well to immunotherapy, highlighting that other currently uncertain biological mechanisms should also be identified and evaluated in the near future [184].

## 8. Biological Samples and Molecular Testing

*KRAS* mutations, notably the *KRAS* G12C mutation, can be identified from tissue samples (bronchial biopsies, transthoracic biopsies, and more rarely biopsies from a metastatic site or a surgically resected sample), from different cytological and fluid samples (bronchial aspirates, fine needle aspiration, endobronchial ultrasound and transbronchial needle aspiration, bronchoalveolar lavage, pleural and cerebrospinal fluid) and from blood [15,105,185,186,187,188,189]. These mutations are currently being characterized using targeted sequencing (RT-PCR and droplet digital PCR) or next-generation sequencing (NGS) approaches [189,190]. Moreover, liquid biopsy is certainly a very promising tool to use at baseline but also at progression in patients treated with specific inhibitors of *KRAS* G12C mutations to track new *KRAS* mutations, notably the Y96D mutation, as well as other mutations such as the R86S and H95D mutations [141]. When considering the increasing number of genes to be evaluated at diagnosis or at tumor progression, NGS must be used as a priority. However, setting up and using NGS systematically can still be associated with a number of constraints, which will be detailed below [191].

## 9. Opportunities and Challenges for the Thoracic Pathologists

Different consensus expert opinions on the use of testing to select treatments targeting *KRAS* inhibitors did not recommend *KRAS*-mutation testing as a daily practice stand-alone assay [13,188]. In recent years, some laboratories have developed *KRAS* testing if *EGFR, ALK*, and *ROS1* genomic alterations are not detected in late-stage NSCLC. However, considering the increasing number of genes of interest for targeted therapy for these tumors, it seems today that this strategy should be abandoned. In this regard, KRAS mutations must be considered with a larger multigene panel, including many other genes of interest, notably *EGFR*, *ALK*, *ROS1*, *BRAF*, *NTRK*, *RET*, *MET*, and *HER2* [13]. In parallel, copy number variation, especially affecting *KRAS,* also needs to be assessed. As discussed above, several genes of interest (such *TP53*, *STK11*, and *KEAP1*) should ideally be concomitantly evaluated in addition to the *KRAS* status [91,104]. At the same time, it is mandatory to assess PD-L1 expression in tumor cells, and if possible, also in immune cells. In this regard, different constraints can arise and present many challenges for a pathology and molecular laboratory (Table 2). Thus, according to the sample size and type, and/or the percentage of tumor cells, the different technologies must be well controlled to make a pathological and a molecular testing report in a short TAT. The accessibility to NGS in daily routine practice and thus assessment of the *KRAS* status is variable from one laboratory to another and from one country to another [191].

### 9.1. Tracking the False-Negative and False-Positive Results

The characterization of the different genomic alterations such as mutations, rearrangements, or amplifications, requires extracted nucleic acid (DNA/RNA) that is not degraded and is obtained in a sufficient quantity for the sequencing method of choice [192]. In this respect, the NGS approaches are becoming more and more sensitive and are now in competition with targeted sequencing methods for use in daily practice. The molecular pathologist needs to select the method of molecular testing depending on the budget and on the hospital organization. Some technologies, such as those based on amplicon-based library preparation, can be more effective than hybrid capture-based NGS library preparation in the case of small tissue lung biopsies and/or a low percentage of tumor cells [174]. The identification of *KRAS* mutations using NGS requires 10–15 ng or 40–50 ng of DNA, respectively, for amplicon-based or hybrid capture technologies [193]. Large panels (more than 300 genes) need a higher amount of DNA than medium-sized panels (from 20 to 50 genes). For multigene panels, *KRAS* is sequenced with deep coverage (>500 x), where the specific depth of coverage depends on whether the assay is a hybrid capture or an amplicon-based approach, as well as whether the values are de-duplicated or not. Globally, the hot spots at codons 12, 13, and 61 can be routinely detected down to at least a 5% variant-allele frequency calling [194].

DNA degradation can provide false-negative or false-positive results, notably in the context of assessment of *KRAS* mutations [167,176]. These false results can be due to poor management of one or several steps of the pre-analytical phase [176,195]. For example, a long period of cold ischemia, a long or short fixative time, can have consequences on false-negative molecular results due to DNA degradation or false-positive molecular results due to DNA deamination [176,195]. 

*KRAS* mutations present on tumor cells must be distinguished from *KRAS* mutations associated with clonal hematopoiesis on indeterminate potential (CHIP) [196,197,198]. CHIP is defined as a “hematologic malignancy-associated somatic mutation” in blood or bone marrow that is not associated with other diagnostic criteria for a hematological malignancy [197,199]. CHIP occurs frequently in the elderly and presents pitfalls for *KRAS* mutation assessment performed with circulating free DNA (cf-DNA) [196,197,198]. In this respect, NGS approaches that employ the simultaneous analysis of cf-DNA and matched DNA blood cells can confirm the presence of somatic mutations and eliminate false identification of the somatic mutation as tumor-specific [197,199]. Moreover, since pancreatic and colon cancers and other solid tumors such as low-grade serous ovarian and endometrial carcinomas show *KRAS* mutations, the presence of this mutation detected with cf-DNA may indicate the co-existence of a carcinoma other than the diagnosed NS-NSCLC [200]. It is noteworthy that a *KRAS* mutation can also be identified in certain non-malignant diseases and theoretically, can be found with cf-DNA, mostly when using very sensitive approaches [201,202,203,204]. For advanced NSCLC, some studies have documented the presence of ct-DNA in about 85% of cases, highlighting that a negative result with a blood sample does not eliminate the presence of the *KRAS* mutation in tissue samples [205]. Thus, naive patients with slow-growing lung cancers may be at risk of having a negative result. In fact, negative results with plasma must also be considered as inconclusive, leading to a tissue biopsy [206].

### 9.2. Algorithm Testing: How to Be Optimal?

Molecular testing for *KRAS* status assessment currently varies according to the country and to the institution and different organizations [207,208,209,210]. If testing follows the international guidelines, NGS needs to be done at diagnosis to test at least *EGFR*, *ALK*, *ROS1*, *BRAF*, *NTRK*, *RET*, and *MET* [13,211]. Therefore, in this context, it seems much easier, more cost-effective, and less tissue-consuming to use an NGS approach than to do multiple RT-PCR analyses with or without immunohistochemistry and fluorescent in situ studies (i.e., for *ALK* and *ROS1*). Considering that other genes of interest (notably *KRAS* and *HER2*) are present in the majority of NGS panels, it is obvious that NGS is the better approach for systematically gaining access to the *KRAS* status for NS-NSCLC [209]. However, NGS at baseline or even at progression and at recurrence is not currently possible in all countries [191,212]. Moreover, one drawback of doing some molecular tests, notably when evaluating the *KRAS* status, is that the therapeutic molecule targeting *KRAS* G12C is not yet available in routine clinical practice and is currently mandatory only for inclusion into clinical trials using KRAS inhibitors. Thus, most of the physicians do not ask for *KRAS* mutation assessments in daily practice. 

We also need to distinguish between bespoke molecular testing, when the pathologist is simply waiting for the physician to request evaluation of the *KRAS* status, and reflex molecular testing, performed by the pathologist for all diagnoses of NS-NSCLC, even without knowing the disease stage at the time of diagnosis and whatever the sample type (tissue biopsies, cytological sample, surgically resected specimen, etc.). It is noteworthy that even if *KRAS* inhibitors are most likely first administered for advanced-stage NSCLC showing a *KRAS* G12C mutation, these drugs will probably be indicated in the future for early-stage NSCLC [213]. The latter strategy will be associated with new challenges for the pathologists, also including the use of NGS in this indication.

### 9.3. Should a Liquid Biopsy Be Integrated into KRAS Status Assessment for NS-NSCLC?

In the absence of tissue or cytological material at diagnosis, the *KRAS* status can be evaluated using a liquid biopsy (LB) [186]. During tumor progression or recurrence, notably in patients treated with different anti-EGFR or anti-ALK TKIs, it is now well-admitted that a LB can be performed systematically to look for different druggable genomic alterations and this can include the detection of a *KRAS* G12C mutation [206]. In case of an inconclusive result, and if possible, depending on the patient status and tumor site, a tissue re-biopsy should be done to track resistance mechanisms [206]. NGS approaches can routinely use a LB and thus provide access to the *KRAS* status and associated genomic alterations [15,190,214,215,216,217]. However, NGS with a LB is currently not often available in most laboratories, in Europe at least, and is set up for routine testing in only a few comprehensive cancer centers [191,215,218]. Therefore, many laboratories use targeted sequencing methods, such as RT-PCR or ddPCR [219,220,221]. Interestingly, a recent study demonstrated that the amount of cf-DNA was higher when some genomic alterations such as *KRAS* were identified in tumors, in contrast to a lower amount of cf-DNA associated with some other mutations detected on *EGFR* or *TP53* [222]. However, the interest in a tissue re-biopsy is currently progressing since a number of mechanisms of resistance (such as gene amplification and rearrangement) are found much more easily or only found (such as histological transformation and epithelial-mesenchymal transformation) in tissue samples and not in blood samples [206]. However, LBs will hopefully be more developed and requested by physicians in the future too, since this non-invasive approach for *KRAS* status evaluation and of other genomic alteration is repeatable and does not need hospitalization for invasive procedures. However, there are still some gaps to fill in this setting in order to better use LBs in daily practice, but this method is certainly also pivotal to look for resistance mechanisms occurring under treatment, in particular with KRAS G12C inhibitors. In this setting, LBs definitively allows an easy monitoring of the disease [218].

### 9.4. To Be Able to Deliver the Results in a Turnaround Time That Follows International Guidelines

As for the other genomic alterations of interest, the TAT to report the *KRAS* status to the physician needs to be mastered in routine clinical practice. According to the international guidelines, this TAT must be no more than 10 working days [13,211]. New considerations have now been set up for reflex NGS panel testing for any newly diagnosed NS-NSCLC [223]. Moreover, this can be performed despite the initial level of information concerning the disease staging status. This option opens great opportunities to considerably reduce the TAT and should enable genotype-directed targeted therapies, including KRAS inhibitors, in an efficient and certainly cost-effective manner, not only for advanced stages, but also in near-treatment strategies for early stages of NSCLC.

### 9.5. Sampling and Timing to Detect the Mechanisms of Resistance Associated with Targeted Therapies of KRAS-Mutated Lung Cancers

As mentioned previously, the biological mechanisms leading to therapeutic resistance to different drugs targeting *KRAS* mutations, notably *KRAS* G12C, can be intrinsic and present at baseline or be acquired at progression under therapy. The issue for the pathologist concerns rapid identification of this mechanism in order to switch the therapy to another molecule [146]. Therefore, the questions that can arise are: (i) what are the best samples for characterization of these mechanisms? and (ii) when should characterizations be done? Some of the resistant mechanisms are certainly easier to assess with a tissue biopsy (such as a *MET* amplification) while other mechanisms may be identified using a liquid biopsy and/or a tissue biopsy (such as new *KRAS* mutations or *BRAF* mutations) [144,145,147]. Moreover, as different mechanisms of resistance occur at progression in patients treated with different TKIs, early detection of this mechanism can be done with blood samples by establishing monitoring, even before the radiological onset of tumor progression [160,161]. However, other mechanisms of resistance, notably histological transformation, can be assessed only with a tissue biopsy.

### 9.6. Other Challenges

*KRAS* mutations are rarely detected in lung squamous cell carcinomas (LSCC) (between 4% to 8%), however, evaluation of the status in LSCC may be required in the near future, resulting in an increase in the workload and an increase in the budget dedicated to molecular biology testing, which should be taken into consideration [74,224].

As previously mentioned, it is very important to combine the status of *KRAS* mutations and other possible biomarkers to better predict the response to immunotherapy. Among the latter, one challenge was implementing the TMB results for patient care, knowing the multiple issues surrounding the use of NGS for TMB evaluation [225,226,227,228]. As an example, the US FDA approved pembrolizumab for the treatment of cancer patients with TMB >10 mutations/megabase (mut/Mb) [229]. Although the approval provides a novel therapeutic option for cancer patients, it raises major concerns [230]. Therefore, a TMB of 10 mut/Mb is certainly an arbitrary cut-off for immunotherapy treatment selection. Additionally, TMB-High, defined by >10 mut/Mb, fails to predict improved ICIs response across different cancer types [230]. Therefore, the reproducibility of TMB cut-off became the focal point for ICI treatment [230]. Thus, despite initially some promising results, the “TMB story” was quite disappointing for many reasons, leading to the fact that this biomarker is not use in routine clinical practice, at least currently in Europe [228]. Conversely, as already mentioned above, to combine the status of *KRAS* and additional biomarkers such as *STK11*, *KEAP1*, and *TP53* is certainly of stronger importance to immunotherapy response prediction [102,103,104].

The assessment and identification of some somatic variants of *KRAS* mutations that are pathogenic (oncogenic) and function could be of strong interest to including patients with these variants into clinical trials [231,232,233]. Along with the development of high-covering NGS panels, the underlying point is whether a NSCLC with an unknown variant of *KRAS* should or should not be included in a precision oncology trial. Well-established pathogenic and benign variants are quite easily recognized, but there is still an urgent need to broaden the classification of a variant of interest from the unknown significance category, from either the potentially benign to the potentially pathogenic (oncogenic) categories, to support the inclusion of a patient in a clinical trial [231,232,233].

The choice of algorithm has a strong impact on the workload of the technicians, the engineers, and the pathologists, including for the post-analytical phase. It is also important to say that most of the molecular testing is usually done on demand by physicians. In this regard, the establishment of reflex testing by the pathologist should be done under a signed agreement with both partners. 

As for other genomic alterations associated with a possible targeted therapy, the different molecular biology tests used in a hospital laboratory for *KRAS* status assessment need to be included in different programs of external quality assurance and obtain accreditation and validation [234,235].

Finally, the pathologists should also set up, in collaboration with the physician, a standby state of the scientific literature concerning different novel technical developments, molecule targets, drug developments, mechanisms of resistance etc., in relation to lung cancer, and more specifically to the different topics concerning *KRAS* in NSCLC. One major challenge will certainly be the ability to rapidly transfer different new biomarkers of responsiveness, resistance and/or toxicity related to the different molecules targeting the *KRAS* mutations to daily practice. This could also be done through different advanced training programs. 

## 10. Perspectives

The current efficacy of the new KRAS G12C inhibitors highlights the urgent need to systematically identify NSCLC patients who can benefit from these therapies alone or in combination with other drugs. In this new situation, pathologists have the responsibility to test for *KRAS* mutations, notably reporting *KRAS* G12C pathogenic variants. This should be done as a routine test using a comprehensive up-front molecular gene panel and before first-line therapy. Accumulated data from preclinical and clinical studies showed that *KRAS* G12C-targeted therapeutics as single agents are inevitably thwarted by drug resistance [143,145,147]. Thus, despite initial promising results, current KRAS specific inhibitors all induce many mechanisms of resistance. Therefore, new developments are ongoing, notably those based on biochemical analyses and some in vitro screening [236,237,238]. A promising strategy to optimize KRAS G12C inhibitor therapy concerns combination treatments with other therapeutic agents, the identification of which is empowered by the insightful appreciation of compensatory signaling pathways or evasive mechanisms, such as those that attenuate immune responses [137,239]. In this regard, many clinical trials are currently open in order to assess efficacy of combination treatments [164]. These new therapeutic strategies associate or will associate different combinations of molecules, including some SHP2 inhibitors or MEK inhibitors together with KRAS G12C specific inhibitors [130,240,241,242,243,244,245]. Other molecules targeting *KRAS* mutations have been tested or are ongoing, such as those targeting mTOR, IGF1R, FASN, or EGLN1 in association with inhibitors targeting *KRAS* G12C mutations, or those that modify glutamine metabolism [246,247]. Moreover, new drugs targeting other *KRAS* subtype mutations, such as *KRAS* G12D, are under evaluation in vitro or in clinical trials [246,248,249]. Pan-KRAS drugs have also the potential to address a broad range of patient populations, including *KRAS* G12D-, *KRAS* G12V-, *KRAS* G13D-, *KRAS* G12R-, and *KRAS* G12A-mutant or *KRAS* wild-type-amplified cancers, as well as cancers with acquired resistance to KRAS G12C inhibitors [246].

## 11. Conclusions

There is a strong necessity to look exhaustively for the different mutations in *KRAS* in NSCLC patients. In this regard, many studies and initiatives are currently ongoing to improve the different aspects related to *KRAS*-mutated lung cancers [250]. However, the histopathologist will continue to play a major role in molecular testing, notably in thoracic oncology [251]. Therefore, the pathologist is more than ever a pivotal actor in this setting and will very soon be at the front line of identification of *KRAS* mutations in NS-NSCLC in routine clinical practice.

## Figures and Tables

**Figure 1 cancers-14-01628-f001:**
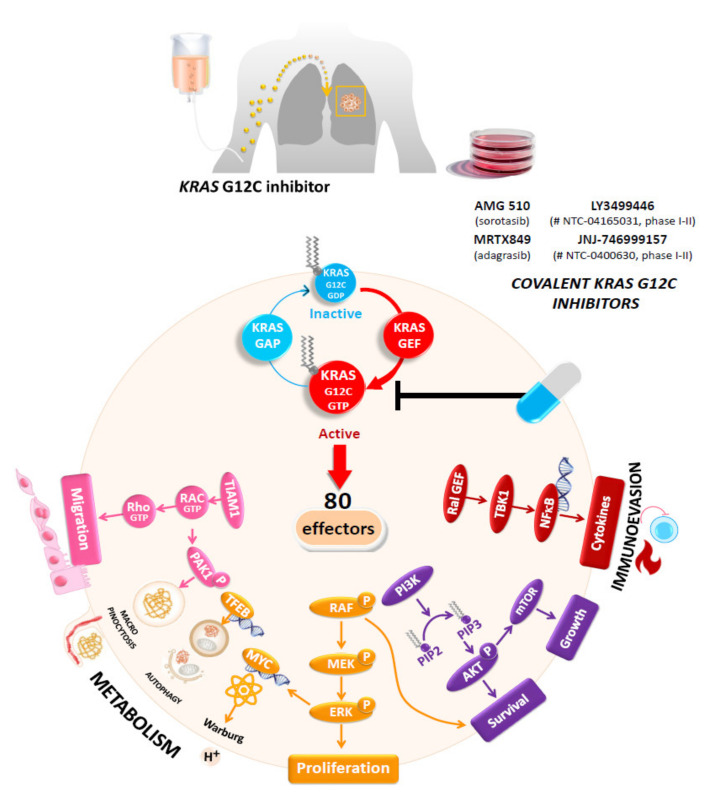
The promise of KRAS G12C inhibitors. Compared with wild-type *KRAS*, cysteine 12 (C12) mutations destroy the GTPase activity of KRAS and lock it in the GTP-bound state. Once activated, Ras signals through 80 effectors, thus activating many different signaling pathways involved in cell proliferation, survival, metabolism, and migration. The best-characterized pathways are the MAPK (Raf-MEK-ERK), PI3K/AKT, and Ral pathways. In contrast, the small molecule drugs sotorasib and adagrasib can form a covalent bond with C12 in the KRAS-G12C protein, causing KRAS to take on an inactive state.

**Figure 2 cancers-14-01628-f002:**
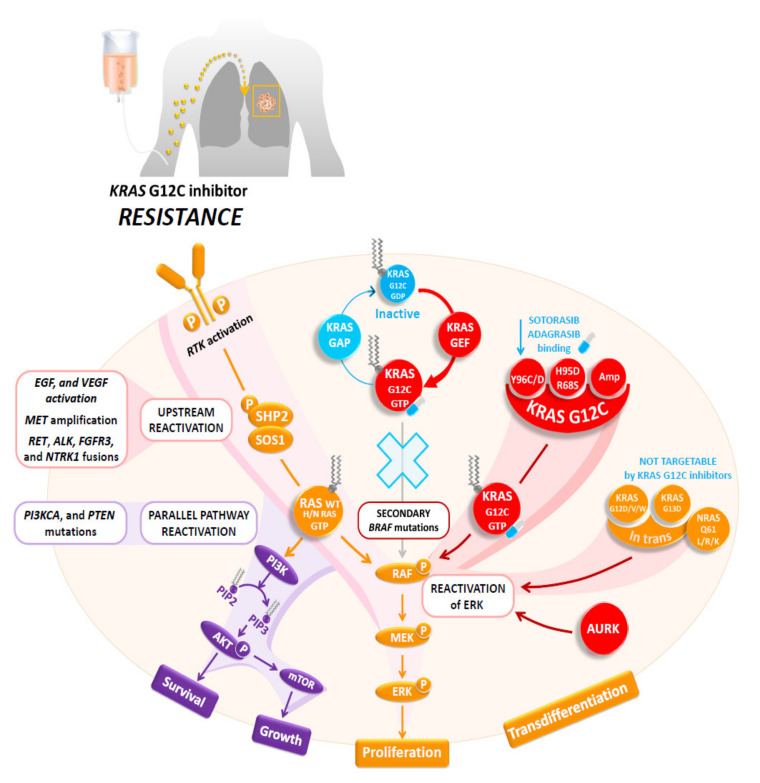
Multifaceted resistance mechanisms to KRAS G12C inhibitors include secondary mutations of *KRAS* G12C that potentiate its nucleotide exchange (Y40A, N116H, or A146V), impair the inherent GTPase activity (A59G, Q61L, or Y64A), or impact other MAPK effectors (such as BRAF mutations). Innate or acquired resistance can also occur through a high RTK activity (EGFR or MET) on the cell surface, which bypasses the KRAS G12C inhibition and activates wild-type N/H-RAS via upstream SHP2 and AURKA activation. Ultimately, this restores the MAPK pathway independently of KRAS G12C and leads to activation of parallel oncogenic pathways such as the PI3K/AKT pathway. Alternatively, *KRAS* amplification or diminished degradation up-regulates KRAS G12C to an excessive level that does not bind the inhibitors.

**Table 1 cancers-14-01628-t001:** A few examples of ongoing clinical trials based on inhibitors targeting *KRAS* in non-small cell lung carcinoma.

Therapeutic Family		
*Clinical trial*	NTC04165031	LY3499446
	NTC0400630	JNJ-746999157
Pan-KRAS inhibitors	NCT04000529	TNO155 + ribocicid or + spartalizumab
	NCT04916236	RMC-4630 + LY3214996
	NCT04111458	BI 1701963 + trametinib
	NCT03114319	TNO155 alone
	NCT04045496	JAB-3312
Dowstream KRAS inhibitors	NCT03681483	RO516766
	NCT03284502	HM95573 + cobimetinib
	NCT02974725	LXH254 + LTT462
	NCT04620330	VS-6766 + defactinib
Upstream + downstream KRAS inhibitors	NCT02230553	Trametinib + lapatinib
	NCT03704688	Trametinib + poniotinib
	NCT04967079	Trametinib + aniotinib
		Futibatinib + binimetinib

**Table 2 cancers-14-01628-t002:** Challenges associated with routine clinical testing of *KRAS* mutations in NSCLC for a pathology laboratory. NGS, next generation sequencing; IHC, immunohistochemistry; IVDR, in vitro drug regulation.

Challenges
To obtain a sufficient quantity and quality of extracted RNA/DNA from formalin fixed tissue biopsies, cytological samples, or liquid biopsies To select the best molecular biology methods (targeted sequencing versus NGS) To integrate the evaluation of genomic alterations of interest (at least on *TP53*, *STK11*, and *KEAP1*) associated with the *KRAS* assessment status To assess the pathogenicity and the functionality of somatic variants of *KRAS* mutations in NSCLC (oncogenicity) To be able to evaluate the PD-L1 status by IHC and the *KRAS* status at the same time to provide in the future immune treatment and/or targeted therapy against a *KRAS* G12C mutation at baseline To deal with possible tumor heterogeneity (according to the size of the sample) To assess the gene *KRAS* status in non-adenocarcinoma lung carcinoma To handle liquid biopsies at baseline and at progression in daily practice To select an optimal gene panel To master the turnaround time required to obtain all molecular biology and IHC results To integrate the costs and the reimbursement according to molecular testing To obtain accreditation according to the ISO 15,189 norm for *KRAS* molecular testing To deal with the next IVDR in Europe To be able to look for mechanisms of resistance at baseline and at progression

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
