# Peer review of "Daily Practice Assessment of KRAS Status in NSCLC Patients: A New Challenge for the Thoracic Pathologist Is Right around the Corner"

_cancers, 2022, doi:10.3390/cancers14071628_

Round 1

Reviewer 1 Report

The authors address the relevant question of integration of KRAS-status analysis in routine pathological workup of NSCLC. They review the available evidence.

L 55-57: Clarify that only approx. 30-50% of KRAS mutations are G12C which is so far the only targetable mutation.

As this is a journal addressing all "cancers, it should be briefly mentioned as an aside which however is gaining clinical significance, that KRAS plays a role in other cancers as well eg. pancreatic for which early positive results with a KRAS G12C inhibitor have recently (ASCO virutal plenary 2/22) been presented.

L 55-57 and 235ff: Also clarify that at least one drug (sotorasib) has already been approved for 2L treatment of KRAS G12C pos. NSCLC.

L 60: KRAS has already been added: Please clarify.

L123: Please correct the dot: of NSCLC.

L235ff: Please add that based on phase II-results (CODEBREAK 100) sotorasib has been approved by FDA and EMA as 2L treatment.

Paragraph 6: Resistance:

Please state that currently resistance seems to occur relatively early (Median around 9 month in various trials, please check and state). This emphasizes the need to identify and address the resistance pathways.

L 306: Please add that trials are ongoing or to start enrolment soon using combination therapy with KRAS-Inh. and 2nd substance targeting a resistance mechanism or a Check-Point-Inhibitor. State some examples which resistance pathways are under investigation as “co-targets” with anti-KRAS-therapy.

L 356f: to my knowledge, no KRAS treatment has been approved as a 1L treatment. Please check.

LL478 ff (LiBi): This paragraph should be rewritten more in favour of liquid biopsy, which certainly will be the future in particular for detection of resistance mechanisms. The argument, that LiBi is not widely available may change in the near future and lead to fewer invasive procedures for patients. However, the considerations in favour of tissue biopsy mentioned in par. 9.5 continue to support tissue re-biopsy if possible. However, in many patients with advanced and progressive NSCLC, a tissue rebiopsy is not possible or realistic, emphasizing the need to develop LiBi as non-invasive tool in this setting.

LL 529 -533: TMB has not gained the clinical relevance attributed to it approx.. 2 – 3 years age. This paragraph should be changed: It is important to assess co-mutations in particular STK11, KEAP, and TP53.

In summary, this is a well-written manuscript on a relevant subject, which in some points (trials addressing primary combination strategies, approval of KRAS-treatment) urgently need updating.

Following required update, the manuscript be fine and of interest to the readers, not only pathologists, but also thoracic oncologists.

Author Response

Responses to reviewer 1

The authors address the relevant question of integration of KRAS-status analysis in routine pathological workup of NSCLC. They review the available evidence.

We wish to thank the reviewer for the very helpful comments and we have now modified our manuscript according to the different suggestions

L 55-57: Clarify that only approx. 30-50% of KRAS mutations are G12C which is so far the only targetable mutation.

We have now clarify this point

As this is a journal addressing all "cancers, it should be briefly mentioned as an aside which however is gaining clinical significance, that KRAS plays a role in other cancers as well eg. pancreatic for which early positive results with a KRAS G12C inhibitor have recently (ASCO virutal plenary 2/22) been presented.

We totally agree with the reviewer that KRAS plays a role in other cancers (notably in colon and pancreatic carcinoma) and that KRASG12C inhibitor could be use in these KRAS G12C mutated tumors. In this regard, the clinical trial [CodeBreaK100 (NCT03600883)] demonstrated the efficacy of the sotorasib in pancreatic carcinoma KRAS G12C mutated. We added these points and some new references as well.

L 55-57 and 235ff: Also clarify that at least one drug (sotorasib) has already been approved for 2L treatment of KRAS G12C pos. NSCLC.

We have now clarify in these paragraphs that sotorasib has been approved for second line treatment of KRAS G12C positive NSCLC.

L 60: KRAS has already been added: Please clarify.

We totally agree with the reviewer and we modified this sentence now

L123: Please correct the dot: of NSCLC.

This typos has been corrected

L235ff: Please add that based on phase II-results (CODEBREAK 100) sotorasib has been approved by FDA and EMA as 2L treatment.

We thank the reviewer for this important point and we have added now this information

Paragraph 6: Resistance:

Please state that currently resistance seems to occur relatively early (Median around 9 month in various trials, please check and state). This emphasizes the need to identify and address the resistance pathways.

We have now better stressed this important point, notably the early onset of resistance under treatment

L 306: Please add that trials are ongoing or to start enrolment soon using combination therapy with KRAS-Inh. and 2nd substance targeting a resistance mechanism or a Check-Point-Inhibitor. State some examples which resistance pathways are under investigation as “co-targets” with anti-KRAS-therapy.

We have added a sentence related to this pivotal point and we detailled some of these therapeutic combination in the the manuscript.

L 356f: to my knowledge, no KRAS treatment has been approved as a 1L treatment. Please check.

We totally agree and we have now modified the sentence accordingly. We added a reference concerning a phase 1 clinical trial (CodeBreaK201) starting recently using the sotorasib as a first line treatment.

LL478 ff (LiBi): This paragraph should be rewritten more in favour of liquid biopsy, which certainly will be the future in particular for detection of resistance mechanisms. The argument, that LiBi is not widely available may change in the near future and lead to fewer invasive procedures for patients. However, the considerations in favour of tissue biopsy mentioned in par. 9.5 continue to support tissue re-biopsy if possible. However, in many patients with advanced and progressive NSCLC, a tissue rebiopsy is not possible or realistic, emphasizing the need to develop LiBi as non-invasive tool in this setting.

It is a very important point and we thank the reviewer. Thus, we modified the paragraph more in favour of liquid biopsy practice now. We stated that this tool will be, hopefully, more developed and requested by the physicians in the future too. We added the fact that there are still some gaps to fill this setting and in order to better use this non-invasive approach in daily practice, but this method is certainly pivotal in order to look for resistance mechanisms occuring under treatment. Moreover this approach definitively allows an easy monitoring of the cancer disease.

LL 529 -533: TMB has not gained the clinical relevance attributed to it approx.. 2 – 3 years age. This paragraph should be changed: It is important to assess co-mutations in particular STK11, KEAP, and TP53.

We totally agree with the reviewer that the TMB story was disapointed and now this biomarker is not use in daily practice despite initially obtained results. So we moderated our statement and we had additional biomarkers of interest that it is certainly more important to combine together (such a STK11, KEAP1 and TP53).

In summary, this is a well-written manuscript on a relevant subject, which in some points (trials addressing primary combination strategies, approval of KRAS-treatment) urgently need updating. Following required update, the manuscript be fine and of interest to the readers, not only pathologists, but also thoracic oncologists.

Reviewer 2 Report

Title: Daily practice assessment of the KRAS status in NSCLC patients. A new challenge for the thoracic pathologist is right around the corner

Authors: Christophe Bontoux, Veronique Hofman, Patrick Brest, Marius Ilie, Baharia Mograbi, Paul Hofman *

Submitted to the section: Molecular Cancer Biology,

Journal: C a n c e r s

Manuscript ID: cancers-1624658

Date: 10 March

Peer-review report: In this review manuscript, Bontoux et. al. have explored the clinical relevance and recent progress made in relevance to KRAS and Non small lung cancer. This is a timely and informative review. Here are my comments:

  1. For the readers of this journal with a broad background, it would benefit if the authors can add a general introduction of Non small lung cancer, its biology, and epidemiological relevance. Refer this section in the manuscript- 'The therapeutic strategy of advanced or metastatic non-small cell lung carcinoma 40 (NSCLC), notably of non-squamous (NS)-NSCLC, has evolved dramatically these last few 41 years, thanks to the development of new targeted therapies and immune checkpoint in- 42hibitors (ICIs) administered alone or in association with chemotherapy and targeted 43 therapies [1-3]'
  2. Refer- 'Duma, N., Santana-Davila, R. and Molina, J.R., 2019, August. Non–small cell lung cancer: epidemiology, screening, diagnosis, and treatment. In Mayo Clinic Proceedings (Vol. 94, No. 8, pp. 1623-1640). Elsevier.' and 'Herbst, R.S., Morgensztern, D. and Boshoff, C., 2018. The biology and management of non-small cell lung cancer. Nature, 553(7689), pp.446-454.'.

3. The Authors can discuss metabolic programming and its perturbation relevant to KRAS driven cancers. Refer - 'Kerk, S.A., Papagiannakopoulos, T., Shah, Y.M. and Lyssiotis, C.A., 2021. Metabolic networks in mutant KRAS-driven tumours: tissue specificities and the microenvironment. Nature reviews Cancer, 21(8), pp.510-525.' https://www.nature.com/articles/s41568-021-00375-9

4. For better clarity,  authors can add a table discussing new promising target KRAS inhibitors.Refer section 'However, caution in interpretation is necessary, until the 243 advantages of overall survival of these direct inhibitors of mutant proteins are confirmed 244 in large-scale phase III randomized trials. In addition, clinical trials are currently being 245 developed to evaluate other KRAS G12C inhibitors such as LY3499446 (clinicaltrials.gov 246 identifier: NTC04165031) and JNJ-746999157 (clinicaltrials.gov identifier: NTC0400630) 247 [120 - 122]. '

Author Response

Responses to reviewer 2

Peer-review report: In this review manuscript, Bontoux et. al. have explored the clinical relevance and recent progress made in relevance to KRAS and Non small lung cancer. This is a timely and informative review. Here are my comments:

  1. For the readers of this journal with a broad background, it would benefit if the authors can add a general introduction of Non small lung cancer, its biology, and epidemiological relevance. Refer this section in the manuscript- 'The therapeutic strategy of advanced or metastatic non-small cell lung carcinoma (NSCLC), notably of non-squamous (NS)-NSCLC, has evolved dramatically these last few years, thanks to the development of new targeted therapies and immune checkpoint in- 42hibitors (ICIs) administered alone or in association with chemotherapy and targetedtherapies [1-3]'

We thank the reviewer for this important suggestion and we added now a general introduction concerning the NSCLC including new references that would be benefit to readers with a broad background

  1. Refer- 'Duma, N., Santana-Davila, R. and Molina, J.R., 2019, August. Non–small cell lung cancer: epidemiology, screening, diagnosis, and treatment. In Mayo Clinic Proceedings (Vol. 94, No. 8, pp. 1623-1640). Elsevier.' and 'Herbst, R.S., Morgensztern, D. and Boshoff, C., 2018. The biology and management of non-small cell lung cancer. Nature, 553(7689), pp.446-454.'

We added these two suggested references in the first part of introduction.

  1. The Authors can discuss metabolic programming and its perturbation relevant to KRAS driven cancers. Refer - 'Kerk, S.A., Papagiannakopoulos, T., Shah, Y.M. and Lyssiotis, C.A., 2021. Metabolic networks in mutant KRAS-driven tumours: tissue specificities and the microenvironment. Nature reviews Cancer, 21(8), pp.510-525.' https://www.nature.com/articles/s41568-021-00375-9

We thank the reviewer for this great suggestion and we added information concerning the metabolic programming and the induced perturbation associated with KRAS driven cancers. In this context, we added now the suggested reference.

  1. For better clarity, authors can add a table discussing new promising target KRAS inhibitors.Refer section 'However, caution in interpretation is necessary, until the advantages of overall survival of these direct inhibitors of mutant proteins are confirmed in large-scale phase III randomized trials. In addition, clinical trials are currently being developed to evaluate other KRAS G12C inhibitors such as LY3499446 (clinicaltrials.gov 246 identifier: NTC04165031) and JNJ-746999157 (clinicaltrials.gov identifier: NTC0400630) 247 [120 - 122]. '

In order to better clarify this section, we have now added a new table concerning some new promising target KRAS inhibitors

Round 2

Reviewer 1 Report

The authors addressed all my concerns. 

I have only a few linguistic remarks:

Abstract

L19: patient’s -> patients’

L 30 just a linguistic remark: Please replace „late-stage” by “advanced-stage” which more common and less judgemental (throughout the manuscript).

Manuscript:

L377: afatinib,

L621: cut -off  cut-off

L623: disappointed – disappointing

L675: look for -> “to assess efficacy of combination treatments”

Author Response

Responses to reviewer 1 (REV2)

The authors addressed all my concerns. 

We wish to thank the reviewer and we have now modified our manuscript according to the different remarks

I have only a few linguistic remarks:

Abstract

L19: patient’s -> patients’

Corrected

L 30 just a linguistic remark: Please replace „late-stage” by “advanced-stage” which more common and less judgemental (throughout the manuscript).

Corrected

Manuscript:

L377: afatinib,

Corrected

L621: cut -off  cut-off

Corrected

L623: disappointed – disappointing

Corrected

L675: look for -> “to assess efficacy of combination treatments”

Corrected